# Exact projected entangled pair ground states with topological Euler invariant

Thorsten B. Wahl [1] ✉, Wojciech J. Jankowski [1], Adrien Bouhon [1,2], Gaurav Chaudhary [1] & Robert-Jan Slager [1,3] ✉

We report on a class of gapped projected entangled pair states (PEPS) with non-trivial Euler topology motivated by recent progress in band geometry. In the non-interacting limit, these systems have optimal conditions relating to saturation of quantum geometrical bounds, allowing for parent Hamiltonians whose lowest bands are completely flat and which have the PEPS as unique ground states. Protected by crystalline symmetries, these states evade restrictions on capturing tenfold-way topological features with gapped PEPS. These PEPS thus form the first tensor network representative of a non-interacting, gapped two-dimensional topological phase, similar to the Kitaev chain in one dimension. Using unitary circuits, we then formulate interacting variants of these PEPS and corresponding gapped parent Hamiltonians. We reveal characteristic entanglement features shared between the free-fermionic and interacting states with Euler topology. Our results hence provide a rich platform of PEPS models that have, unexpectedly, a finite topological invariant, forming the basis for new spin liquids, quantum Hall physics, and quantum information pursuits.

Tensor network states (TNS) form a generally applicable tool for the description of quantum matter. A numerically efficient representation of the ground states of local gapped Hamiltonians[1–5], TNS play a pivotal role both in the simulation of correlated systems[6–11] and the analytical classification of topological phases[12–17]. Yet, due to the increased complexity in higher dimensions, TNS have not yet matched the success of non-interacting band theory[18,19], both in simulations and the classification of topological phases[20–26]. Of particular interest are therefore systems which can be well captured with band theory but are marked by difficulties when it comes to TNS approaches. The most well-known such example is chiral topological systems: In topological band theory, they are characterized by occupied bands whose overall Chern number is non-vanishing, separated by a gap from the conduction bands. While TNS approaches are able to capture the chiral topological features of such systems, this comes at the cost of producing algebraically decaying correlations characteristic of critical systems[27–29]. Generally, it has been shown that TNS with exponentially decaying correlations cannot capture any higher-dimensional topological invariant[30] of the ten Altland-Zirnbauer (AZ) classes[31].

The severe restrictions of TNS to represent gapped non-interacting topological phases suggest that TNS might equally struggle to capture topological phases protected by crystalline symmetries. However, reinvigorated interests[32,33] in relation to quantum geometry[34,35] could provide a useful tool in that they outline flatband conditions. Under such conditions, it is possible to define topological flatband Hamiltonians that can be formulated as exemplary parent Hamiltonians. These are sums of projectors with local support that each annihilate the ground state(s). From these local projectors, a TNS ground state can in principle be constructed. However, whether the resulting state is non-vanishing and can be made the unique ground state of such a crystalline symmetry-protected topological Hamiltonian might be hampered by the previously mentioned hurdles. Intuitively, TNS with exponentially decaying correlations are incompatible

[1]TCM Group, Cavendish Laboratory, Department of Physics, Cambridge, UK. [2]Nordita, Stockholm University and KTH Royal Institute of Technology, Stockholm, Sweden. [3]Department of Physics and Astronomy, University of Manchester, Oxford Road, Manchester M13 9PL, United Kingdom. ✉e-mail: tw344@cam.ac.uk; rjs269@cam.ac.uk

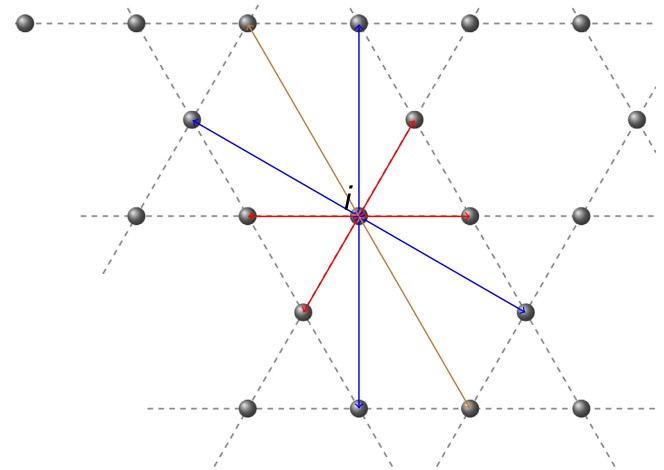

**Fig. 1 | Hoppings realized by the model Hamiltonian [Eq. (2)].** nearest-neighbor (red), next-nearest-neighbor (blue), and third-nearest-neighbor within hexagons (brown) from site $i$.

with topological invariants, as they come with delocalized edge modes around a physical boundary (in more than one dimension); the local structure of tensor networks is incapable of separating such edge modes from the bulk modes, delocalizing them as well. Because of that, in the following, we consider crystalline symmetry-protected topological phases which do not have helical or chiral edge modes.

We show that a family of topological projected entangled pair states (PEPS)[36] and generating Hamiltonians can be formulated in the context of the Euler class[25,37–39]. The Euler class is a multi-gap invariant[25], pertaining to topological structures that emerge when groups of partitioned bands (band subspaces) carry non-trivial topological indices[25]. These topological charges of groups of bands can be altered by braiding nodes in momentum space, as band nodes residing between neighboring bands can carry non-Abelian charges[37,38,40]. The braiding of non-Abelian frame charges and multi-gap topologies have been increasingly related, both theoretically and experimentally, to physical systems that range from out-of-equilibrium settings[41–43] and phonon spectra[44,45], to electronic systems (twisted, magnetic and conventional)[38,46,47] as well as metamaterials[48–50].

The here introduced family of PEPS has non-trivial Euler class, evading no-go conditions as for tenfold-way topologies[31], constituting a 2D analog of the Kitaev chain[51]. These PEPS differ from the higher-order topological insulators represented by PEPS in Ref. 52, which have a bond dimension in one spatial direction that grows with the system size. Within the non-interacting limit, from a band theory perspective, our PEPS enjoy ideal quantum geometrical properties, which we elucidate in the Supplementary Information (SI) in detail. More importantly, by applying shallow quantum circuits of diagonal unitaries, we transform these PEPS and their gapped parent Hamiltonians to interacting variants. We can signify the Euler phase both in the non-interacting and the interacting limit upon appealing to the entanglement spectrum. As such, our results set a benchmark for an exact class of PEPS parent Hamiltonians with finite topological invariant.

## Results

### Euler class
A pair of isolated (gapped from the rest of the spectrum) bands $|u_n(\boldsymbol{k})\rangle$, $|u_{n+1}(\boldsymbol{k})\rangle$ can acquire a non-trivial Euler class $\chi$ when it is part of at least a three-band system that enjoys a reality condition assured by the presence of $C_2\mathcal{T}$ [twofold rotations combined with time-reversal symmetry (TRS)], or $\mathcal{PT}$ symmetry, involving parity and TRS. The Euler

class is then concretely obtained as[37,38]

$$\chi = \frac{1}{2\pi}\int_{\mathrm{BZ}} \mathrm{Eu}\, dk_1 \wedge dk_2 \in \mathbb{Z}, \tag{1}$$

where one integrates the Euler curvature $\mathrm{Eu} = \langle \partial_{k_1} u_n(\boldsymbol{k})|\partial_{k_2} u_{n+1}(\boldsymbol{k})\rangle - \langle \partial_{k_2} u_n(\boldsymbol{k})|\partial_{k_1} u_{n+1}(\boldsymbol{k})\rangle$ over the Brillouin zone (BZ). The pair of bands can either be degenerate (and flat) or feature a number of $2\chi$ nodal points that cannot be annihilated due to the topological nature[25,37,38]. Eq. (1) shows that the Euler class is the real analog of the Chern number. Similarly, the isolated two-band subspace does not admit exponentially-localized Wannier functions in a $C_2\mathcal{T}$-symmetric gauge, but unlike the Chern case, the system does not feature protected chiral or helical edge states, allowing for a PEPS representation. Our PEPS construction will break the Wannier restriction to $C_2\mathcal{T}$ symmetric gauge and three bands; more precisely, the latter, as it starts with virtual particles occupying six bands. The projection from virtual to physical fermions reduces the number of bands back to three, constituting a (non-invertible) map between the gauge-symmetry breaking input state and the gapped Euler state. Hence, a representation by a PEPS with a finite bond dimension is made possible by the PEPS construction breaking the gauge symmetry.

### The model
To concretize the discussion, we consider spinless fermions hopping on the kagome lattice with nearest-neighbor hopping $t = -1$, next-nearest-neighbor hopping $t' = -1$ and third-nearest-neighbor hopping $t'' = -1$ inside the hexagons (see Fig. 1). For chemical potential $\mu$, the Hamiltonian thus reads

$$H = \sum_{\langle i,j\rangle} a_i^\dagger a_j + \sum_{\langle\langle i,j\rangle\rangle} a_i^\dagger a_j + \sum_{\langle\langle\langle i,j\rangle\rangle\rangle_{\bigcirc}} a_i^\dagger a_j - \mu\sum_{i=1}^{N} a_i^\dagger a_i, \tag{2}$$

where $\langle i,j\rangle$, $\langle\langle i,j\rangle\rangle$ correspond to nearest- and next-nearest neighbor pairs of sites $i, j$ and $\langle\langle\langle i,j\rangle\rangle\rangle_{\bigcirc}$ to third-nearest neighbor pairs of the same hexagons. $a_i^\dagger$ ($a_i$) are the fermionic creation (annihilation) operators and $N$ the number of sites. The Hamiltonian has two degenerate flat bands at $E = -2 - \mu$ and a dispersive band on top, separated by an energy gap $\Delta = 3$; for more details on the model, see Methods. The flat bands have Euler number $\chi = 1$, protected by $C_2\mathcal{T}$ symmetry. At $\mu = -2$, both flat bands are at $E = 0$, and the ground states $|\psi_n\rangle$, $n = 0, 1, \ldots, 2N/3$ are macroscopically degenerate, characterized by fillings $[0, 1, 2, ..., 2N/3]$. We now construct the ground state with the highest filling, which will become the unique ground state for $-2 < \mu < 1$. To that end, we note that the Hamiltonian for $\mu = -2$ can be rewritten as

$$H_p = \sum_{\bigcirc}\sum_{i,j\in\bigcirc} a_i^\dagger a_j = 6\sum_{\bigcirc} a_{\bigcirc}^\dagger a_{\bigcirc} = 6\sum_{\bigcirc} h_{\bigcirc}, \tag{3}$$

where $\bigcirc$ denotes the hexagons of the kagome lattice, and we defined $a_{\bigcirc} = \frac{1}{\sqrt{6}}\sum_{i\in\bigcirc} a_i$ and $h_{\bigcirc} = a_{\bigcirc}^\dagger a_{\bigcirc}$. Hence, the ground states fulfill $a_{\bigcirc}|\psi_n\rangle = 0$ for all hexagons $\bigcirc$. The ground state with the highest occupation number is

$$\left|\psi_{2N/3}\right\rangle = \prod_{\bigcirc} a_{\bigcirc}|1\ldots 1\rangle, \tag{4}$$

where $|1\ldots 1\rangle$ is the fully occupied state. Due to $\{a_{\bigcirc}, a_{\bigcirc'}\} = 0$, the ordering in Eq. (4) is irrelevant. However, notably, for the other commutation relations, we have $\{a_{\bigcirc}, a_{\bigcirc'}^\dagger\} = \delta_{\bigcirc,\bigcirc'} + \frac{1}{6}\delta_{\langle\bigcirc,\bigcirc'\rangle}$,

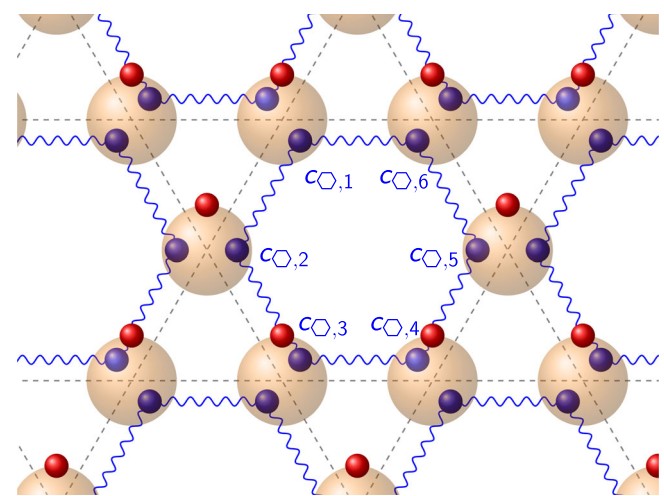

**Fig. 2 | Projected entangled simplex state.** The blue wiggly lines denote the initial state of virtual fermions $c_{\bigcirc,k}$ (blue balls) entangled across hexagons. The transparent red balls denote the projection onto the physical fermions (red balls).

where $\langle \bigcirc, \bigcirc' \rangle$ denotes corner-sharing, neighboring hexagonal plaquettes. This anticommutation algebra shows that, while the operators $a_{\bigcirc}^{\dagger}$ effectively create fermions in a superposition of six atomic orbitals, the Hamiltonian Eq. (3) is not adiabatically connected to a Hamiltonian of an atomic insulator, despite the functional similarity to such Hamiltonians: In particular, even if one deforms our quasiparticle operators $a_{\bigcirc}$ (see following section), their non-trivial anticommutation properties are protected by $\mathcal{C}_2 \mathcal{T}$ symmetry. That is, the $a_{\bigcirc}$ cannot be deformed into single-site operators (and similarly for the Hamiltonian) without breaking $\mathcal{C}_2 \mathcal{T}$ symmetry.

$|\psi_{2N/3}\rangle$ can be written as a projected entangled simplex state (PESS)[53] as follows: We start out with a virtual state of $2N$ spinless fermions – two assigned to each physical fermion. The virtual fermions are in the state $|\omega_v\rangle = \frac{1}{\sqrt{6}} \prod_{\bigcirc} \sum_{i=1}^{6} c_{\bigcirc,i} |1_v\rangle$, where $|1_v\rangle$ denotes the fully occupied virtual state. $c_{\bigcirc,i}$ corresponds to virtual fermion $i = 1, ..., 6$ within hexagon $\bigcirc$ (as opposed to the physical particles $a_j$, there are now unique assignments within hexagons), cf. Fig. 2. The next step is to map each pair of virtual fermions around a site to one physical fermion. To that end, we use the operator $\hat{M}_j = a_j^{\dagger} c_j' c_j + c_j' - c_j$. Here, $c_j'$ corresponds to the virtual fermion located on the left of the site $j$ and $c_j$ to the virtual fermion located on its right. We finally project on the vacuum of virtual particles, obtaining the overall state

$$|\psi_{\text{PEPS}}\rangle = \langle 0_v| \prod_{j=1}^{N} \hat{M}_j \prod_{\bigcirc} \frac{1}{\sqrt{6}} \sum_{i=1}^{6} c_{\bigcirc,i} |1_v 0_p\rangle, \quad (5)$$

where $|1_v 0_p\rangle$ corresponds to the vacuum of physical fermions and fully occupied virtual fermionic state. We already labeled the overall state as a "PEPS", since it can also be written as the more familiar projected entangled pair state, as we show further below. We note that it is a fermionic PEPS where tensors are replaced by fermionic operators[54]. The map $\hat{M}_j = a_j^{\dagger} c_j' c_j + c_j' - c_j$ has the following effect on the superposition of basis states in $\prod_{\bigcirc} \frac{1}{\sqrt{6}} \sum_{i=1}^{6} c_{\bigcirc,i} |1_v 0_p\rangle$: If site $j = \bigcirc \cap \bigcirc'$ is neither affected by $c_i$ with $i \in \bigcirc$ nor $c_k$ with $k \in \bigcirc'$ (i.e., there are two virtual fermions at site $j$), then a physical fermion is created via $a_j^{\dagger}$. If site $j$ is affected by either $c_{i\in\bigcirc}$ or $c_{k\in\bigcirc'}$ (i.e., there is one virtual fermion at site $j$), then no physical fermion is created at site $j$, leaving it in the physical vacuum state. If site $j$ is affected by both $c_{i\in\bigcirc}$ and $c_{k\in\bigcirc'}$ (i.e., there is no virtual fermion at site $j$), the basis state is annihilated. The negative sign in $\hat{M}_j$ is necessitated by the fermionic anticommutation relations in the PEPS construction.

In order to demonstrate that $|\psi_{2N/3}\rangle \propto |\psi_{\text{PEPS}}\rangle$, we first verify that $a_{\bigcirc} |\psi_{\text{PEPS}}\rangle = 0$ and later that the PEPS has filling $2N/3$. For the first claim, we notice that

$$
\begin{aligned}
&\langle 0_v| a_j(a_j^{\dagger} c_j' c_j + c_j' - c_j)[\ldots]|1_v 0_p\rangle \\
&= \langle 0_v|(a_j^{\dagger} c_j' c_j + c_j' - c_j) c_j'[\ldots]|1_v 0_p\rangle \\
&= \langle 0_v|(a_j^{\dagger} c_j' c_j + c_j' - c_j) c_j[\ldots]|1_v 0_p\rangle,
\end{aligned} \quad (6)
$$

where [...] denotes a sum of products of operators that do not act on the physical fermion at site $j$. We thus have

$$
\begin{aligned}
a_{\bigcirc'} |\psi_{\text{PEPS}}\rangle = &\pm \frac{1}{6} \langle 0_v| \prod_j \left( a_j^{\dagger} c_j' c_j + c_j' - c_j \right) \sum_{i=1}^{6} c_{\bigcirc',i} \times \\
&\times \prod_{\bigcirc} \sum_{k=1}^{6} c_{\bigcirc,k} |1_v 0_p\rangle = 0.
\end{aligned} \quad (7)
$$

Second, we see that the initial state $|1_v 0_p\rangle$ contains $2N$ virtual and no physical fermions. The operator $\prod_{\bigcirc} \sum_{k=1}^{6} c_{\bigcirc,k}$ reduces that to $2N(1 - 1/6) = 5N/3$ fermions. Finally, each operator $\hat{M}_j$ creates one physical fermion less than it annihilates virtual ones, i.e., we are left with $2N/3$ physical fermions in $|\psi_{\text{PEPS}}\rangle$. Hence, $|\psi_{\text{PEPS}}\rangle \propto |\psi_{2N/3}\rangle$, as claimed.

The PESS we have considered so far can be converted into a PEPS by realizing that the simplex states are of the form $|011111\rangle + |101111\rangle + \ldots + |111110\rangle$, also known as a $W$-state[55], which can be written as a non-translationally invariant matrix product state of bond dimension 2 or a translationally invariant one of bond dimension 6. $\hat{M}$ can be represented as a rank-3 tensor $M_{ab}^i$ with $M_{11}^1 = M_{10}^0 = -M_{01}^0 = 1$ and all other elements equal zero. The resulting PEPS tensor has rank 5 and bond dimension 2 or 6, respectively, see Fig. 3. We note that a similar construction in terms of PESS defined on triangles can be used to describe the ground state of the Hamiltonian (2) for nearest-neighbor hopping only, an Euler system with one flat bottom band touched by two dispersive bands from above[48,49].

We now show that the PEPS is the unique ground state for $-2 < \mu < 1$: For chemical potential $\mu = -2$, the ground state subspace $S$ is the intersection of all null spaces of $h_{\bigcirc}$, i.e., $S = \text{span}\{\prod_{\bigcirc} a_{\bigcirc} a_{\bigcirc}^{\dagger} |n\rangle\}_n = \text{span}\{\prod_{\bigcirc} a_{\bigcirc} |n\rangle\}_n$, where $\{|n\rangle\}_n$ is a complete basis and we used that $h_{\bigcirc}^{\perp} = a_{\bigcirc} a_{\bigcirc}^{\dagger}$ is a projector onto the orthogonal complement of $h_{\bigcirc}$. Each $a_{\bigcirc}$ eliminates one fermion from the basis state $|n\rangle$. Hence, the state with the highest occupation in $S$ is $\prod_{\bigcirc} a_{\bigcirc} |1 \ldots 1\rangle$, and it has filling $N - N/3 = 2N/3$. Any other state contained in $S$ has lower expectation value of the overall occupation. For $-2 < \mu < 1$, this highest occupation ground state becomes the unique ground state, as all other states contained in $S$ have lower overall occupation expectation value and thus get penalized. This example shows that PEPS can be the unique ground states of local gapped Hamiltonians with non-trivial two-dimensional crystalline topological features, even in the non-interacting limit. This contrasts with the inability of free-fermionic PEPS to capture any higher-dimensional topological labels of the ten-fold classification unless the PEPS have algebraically decaying correlations[27,28,31,56].

### Free-fermion generalizations
A straightforward generalization is obtained by modifying the simplex states to a linear combination, $|\tilde{\omega}_v\rangle = \prod_{\bigcirc} \sum_{i=1}^{6} \beta_i c_{\bigcirc,i} |1_v\rangle$ with $\beta_i \in \mathbb{C}$, and keeping $\hat{M}_j$ the same. The new PEPS is annihilated by $\tilde{a}_{\bigcirc} = \sum_{i\in\bigcirc} \beta_i a_i$, where we set $\sum_{i=1}^{6} |\beta_i|^2 = 1$. This corresponds to the

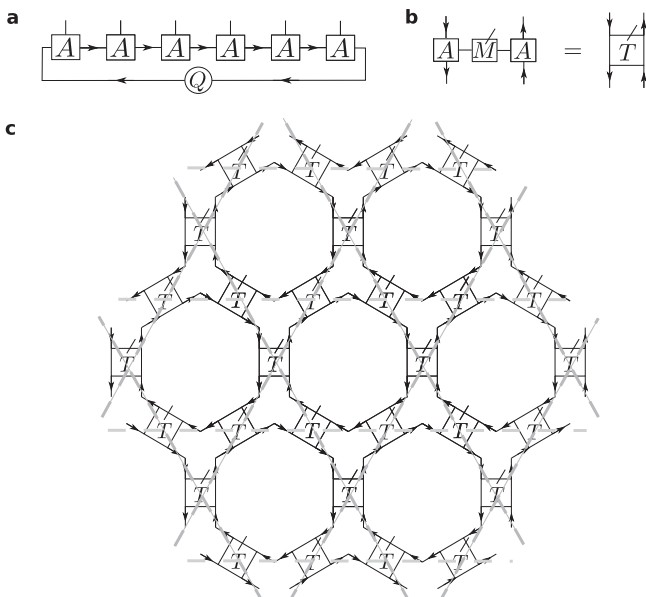

**Fig. 3 | PEPS construction. a** Matrix product state representation of the simplex states residing on the hexagons. $A$ can either be chosen to be of bond dimension $D = 2$, with $A_{12}^0 = 1/\sqrt{6}$, $A_{11}^1 = -A_{22}^1 = 1$, $Q_{21} = 1$ and all other elements of $A$ and $Q$ equal to zero, or $D = 6$ with $A_{61}^0 = 1/\sqrt{6}$, $A_{l,l+1}^1 = 1$ ($l = 1, ..., 5$), all other elements of $A$ equal to zero and $Q = \mathbb{1}$ (translationally invariant representation). Incoming arrows denote left and outgoing arrows right lower indices. **b** By combining two $A$ tensors with the tensor $M$, we obtain the tensor $T$ constituting the PEPS. **c** PEPS with one rank-5 tensor located on each site of the kagome lattice (gray dashed lines).

new Hamiltonian

$$\tilde{H} = 6 \sum_{\bigcirc} \sum_{i,j \in \bigcirc} \beta_i^* \beta_j a_i^\dagger a_j - (\mu + 2) \sum_{i=1}^{N} a_i^\dagger a_i. \quad (8)$$

One can check that $\mathcal{C}_2\mathcal{T}$ symmetry implies $\beta_{i+3}^* = \beta_i$ ($i = 1, 2, 3$) up to an irrelevant overall phase. For any choice of $\{\beta_i\}_{i=1}^{6}$, the corresponding PEPS is an Euler insulator, as the corresponding parent Hamiltonians are always gapped and continuously connected to the original one. Correspondingly, we have $\{\tilde{a}_{\bigcirc}, \tilde{a}_{\bigcirc'}^\dagger\} = \delta_{\bigcirc,\bigcirc'} + |\beta_{\bigcirc \cap \bigcirc'}|^2 \delta_{\langle\bigcirc,\bigcirc'\rangle}$, where there are always $i = \bigcirc \cap \bigcirc'$ for which the final term is non-vanishing.

### Interacting generalizations

The modified simplex states give rise to non-interacting PEPS. We present two ways of generalizing these PEPS to the interacting regime. We will only briefly consider the first one, which employs a Gutzwiller projection and is likely to give rise to a new type of symmetry-enriched topological order[57]. The second example we consider applies symmetry-preserving shallow quantum circuits on the non-interacting PEPS. Therefore, it is guaranteed that the resulting interacting PEPSs are in the same topological phase as the original non-interacting one[58].

To obtain the first type of interacting Euler state, we apply a Gutzwiller projection $P_G = \prod_i (a_{i\uparrow}^\dagger a_{i\uparrow} a_{i\downarrow} a_{i\downarrow}^\dagger + a_{i\uparrow} a_{i\uparrow}^\dagger a_{i\downarrow}^\dagger a_{i\downarrow})$ on two copies (denoted by $\uparrow$ and $\downarrow$) of the non-interacting PEPS, $|\psi_{\text{PEPS}}^\uparrow\rangle$ and $\mathcal{PH}|\psi_{\text{PEPS}}^\downarrow\rangle$, where $\mathcal{PH}$ induces a particle-hole transformation. This is necessary, as the Gutzwiller projector $P_G$ enforces one fermion per site, i.e., in order for it not to annihilate the state, we need overall filling fraction 1 in the two PEPS copies. The resulting state

$$|\psi_{\text{PEPS}}^{S=1/2}\rangle = P_G |\psi_{\text{PEPS}}^\uparrow\rangle \mathcal{PH}|\psi_{\text{PEPS}}^\downarrow\rangle \quad (9)$$

is expected to be an Euler quantum spin liquid and to have fractional statistics, similarly to the Gutzwiller projection of two $p + ip$ superconducting states[59]. As the symmetries remain preserved under the Gutzwiller projection, $|\psi_{\text{PEPS}}^{S=1/2}\rangle$ is expected to be a symmetry-enriched topologically ordered state. This implies topological ground state degeneracy on the torus and anyonic excitations that survive breaking of the symmetry. If the protecting symmetry is not broken, additional features of non-interacting Euler insulators are expected, such as the cusp in the entanglement spectrum at $K = 0$ described below. However, studying the precise physical properties of this state goes beyond the scope of our work.

Second, we can also generalize the construction to interacting states by applying a shallow quantum circuit $U$ of diagonal unitaries on the non-interacting PEPS, which makes it easy to ensure that $\mathcal{C}_2\mathcal{T}$ symmetry is preserved. Hence, by definition, we remain in the same topological phase. Furthermore, the new state will also be a PEPS of low bond dimension. We consider the simplest case of nearest-neighbor gates. We view these as being applied on all hexagons in a translationally invariant fashion. Within each hexagon, we label $u_{j,j+1}$ as the unitary acting on sites $j$ and $j+1$ ($j = 7 \equiv 1$) inside a given hexagon, with sites enumerated as in Fig. 2. $\mathcal{C}_2\mathcal{T}$ symmetry is achieved if $u_{j,j+1} = u_{j+3,j+4}^*$ for all $j = 1, 2, 3$. The simplest continuously tuneable case is $u_{j,j+1} = \mathbb{1} - (1 - e^{\pm i\alpha})n_j n_{j+1}$ with particle number operators $n_j$, $\alpha \in [0, 2\pi)$, and positive (negative) sign for $j = 1, 2, 3$ ($j = 4, 5, 6$). The new PEPS is given by $|\psi'_{\text{PEPS}}\rangle = U|\psi_{\text{PEPS}}\rangle$ and the Hamiltonian gets transformed as

$$H' = UHU^\dagger = \sum_{\bigcirc} \sum_{i,j \in \bigcirc} a_i'^\dagger a_j' - (\mu + 2) \sum_{i=1}^{N} a_i^\dagger a_i, \quad (10)$$

where we defined $a_i' = U a_i U^\dagger$ and used that $U$ commutes with $n_i = a_i^\dagger a_i$. One can easily verify $a_i'^\dagger = a_i^\dagger \prod_j^{\langle i,j \rangle}[\mathbb{1} - (1 - e^{i\sigma_{\langle i,j \rangle}\alpha})n_j]$, where the product runs over all nearest neighbors of site $i$. $\sigma_{\langle i,j \rangle} = +1$ if $\langle i, j \rangle$ corresponds to one of the first three bonds in the hexagon that it lies in and $-1$ if it corresponds to one of the last three bonds. This gives rise to the overall Hamiltonian

$$H' = \sum_{\bigcirc} \sum_{i,j \in \bigcirc} a_i^\dagger \prod_k^{\langle k,i \rangle}[\mathbb{1} - (1 - e^{i\sigma_{\langle k,i \rangle}\alpha})n_k] \times$$
$$\times \prod_l^{\langle j,l \rangle}[\mathbb{1} - (1 - e^{-i\sigma_{\langle j,l \rangle}\alpha})n_l] a_j - (\mu + 2) \sum_{i=1}^{N} n_i. \quad (11)$$

$H'$ has the same spectrum as $H$ for fixed $\mu$, and $|\psi'_{\text{PEPS}}\rangle$ is therefore its unique ground state for $-2 < \mu < 1$. The Hamiltonian is strictly local, acting on hexagons $\bigcirc$ and adjacent triangles. Its four-body interactions have amplitude $\mathcal{O}(\alpha)$ and higher-body interactions are of higher order. This is the first example of an interacting Euler insulator with a local gapped Hamiltonian. $|\psi'_{\text{PEPS}}\rangle$ can be constructed by writing the phase matrix of $u_{j,j+1} = \mathbb{1} - (1 - e^{\pm i\alpha})n_j n_{j+1}$ as $\sum_{q=1}^{2} R_q^{ab} R_q^{cd}$ with $R_1^{ab} = \delta_{ab}$ and $R_2^{ab} = \sqrt{-1 + e^{\pm i\alpha}}\delta_{1a}\delta_{1b}$, $a, b \in \{0, 1\}$. (As the underlying operators are even, they can be decomposed into tensor products.) Four nearest-neighbor unitaries act on each site, such that the tensors of $|\psi'_{\text{PEPS}}\rangle$ can be constructed by contracting the physical leg of $T$ with four $R$-tensors, see Fig. 4. If the bond dimension of $T$ was chosen to be 2, the interacting PEPS has bond dimension $D = 4$. We note that even though the protection of Euler topology under $\mathcal{C}_2\mathcal{T}$ symmetry in the presence of interactions is an open problem, in our case, we can guarantee that Euler topology is preserved, as our interacting PEPS $|\psi'_{\text{PEPS}}\rangle$ and interacting model Hamiltonian are both related via a symmetry-preserving shallow quantum circuit to their non-interacting counterparts. A shallow quantum circuit does not change topological features, as these are global[58]. Hence, we are in the same symmetry-protected topological phase for all $\alpha \in [0, \pi]$. This can also be seen

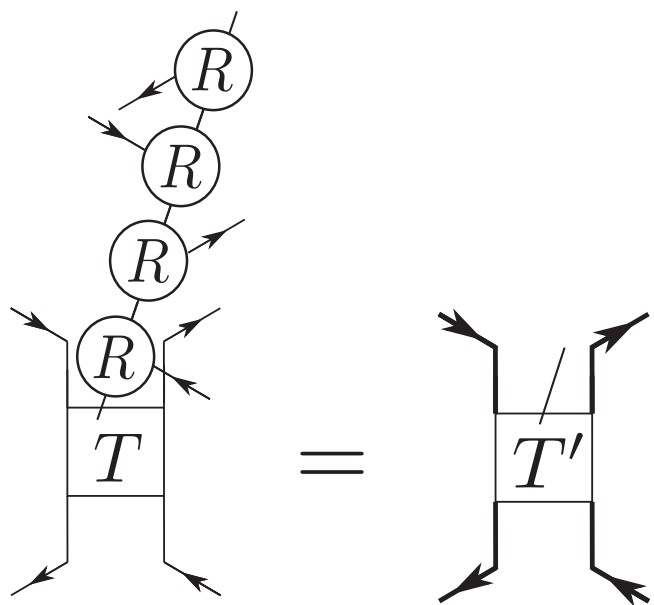

**Fig. 4 | Construction of the tensors $T'$ forming the building blocks of the interacting $|\psi'_{PEPS}\rangle$.** The $R$-tensors get absorbed into the $T$ tensor, increasing its bond dimension (indicated by thick directed lines).

from the fact that the Hamiltonians $H'$ have the same energy spectrum (and gap) for all $\alpha$.

### Entanglement spectra

We numerically calculated the entanglement spectra of $|\psi'_{PEPS}\rangle$ for an infinitely long torus with circumference $L_y$. That is, the torus is bipartitioned with $L_y$ unit cells located around the perimeter of the resulting cylinder. We obtained its entanglement spectrum by calculating the non-interacting $|\psi_{PEPS}\rangle$ using TenPy[60] and applying the quantum circuit $U$ on it to obtain the interacting $|\psi'_{PEPS}\rangle$. The entanglement spectra for various values of $\alpha$ and $L_y = 6$ are shown in Fig. 5. We observe that the low-lying part of the entanglement spectrum possesses a cusp at momentum $K = 0$, which remains intact as $\alpha$ is increased: Our non-interacting model corresponds to two degenerate Chern bands with opposite Chern number $C = \pm 1$. Due to continuous connection to the non-interacting limit, we expect that our $K = 0$ mode in the interacting case is not isolated, but connected to two isolated branches symmetric around the $K = 0$ axis (gapless counter-propagating modes). Revealing these branches would require vastly larger system sizes, which are beyond the scope of our work. We further detail the entanglement features of the non-interacting case, including the stable cusp at $K = 0$, in the Methods.

## Discussion

We leverage quantum geometric conditions (see Methods and SI) to define a class of exact PEPS with finite topological Euler invariant. The enigmatic nature of the Euler class allows to circumvent no-go conditions. Importantly, these models can be generalized to interacting variants and have definite entanglement signatures. As such, these PEPS set a benchmark for new pursuits. These potential pursuits involve studying exotic excitations and spin liquids realized from Euler many-body PEPS ground states. In particular, on introducing interactions, novel kinds of fractionalizations should emerge from the interplay of the many-body entanglement as well as emergent quantum anomalous Hall states[61,62]. In addition, as all our states can be created by shallow quantum circuits from product states and have topological features, they are also particularly interesting for implementations on noisy intermediate-scale quantum devices and the development of

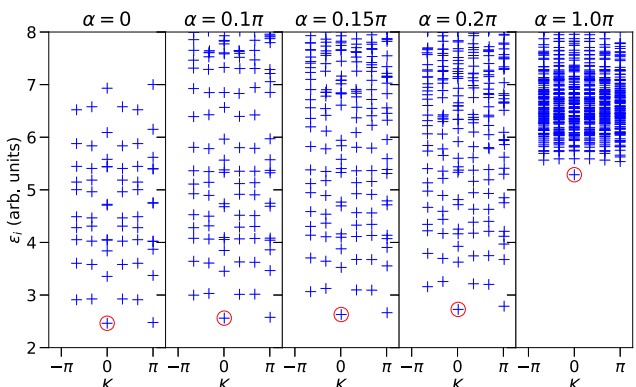

**Fig. 5 | Entanglement spectra as a function of the many-body momentum $K$ for different values of $\alpha$ for $L_y = 6$.** For small values of $\alpha$, the low-lying spectrum strongly resembles the non-interacting one ($\alpha = 0$). In particular, a cusp at $K = 0$ (highlighted by a red circle) is preserved as $\alpha$ is increased. Parallel to this, new entanglement energies appear at the top of the spectrum and eventually merge with its low-lying part.

new quantum error correction protocols. We will report on this in the near future.

## Methods

### Momentum-space characterization of the model in the non-interacting limit

We demonstrate how the model introduced in the main text, Eq. (2), can be decomposed in momentum space. Furthermore, in the SI, we showcase the ideal non-Abelian quantum geometry realized in the topological Euler bands in the non-interacting limit.

We first Fourier transform the real-space creation and annihilation operators to the basis of Bloch orbitals: $\tilde{a}^{\dagger}_{\alpha,\mathbf{k}} = \frac{1}{\sqrt{N}}\sum_l a^{\dagger}_{\alpha,l} e^{-i\mathbf{k}\cdot(\mathbf{R}_l + \mathbf{r}_\alpha)}$, $\tilde{a}_{\alpha,\mathbf{k}} = \frac{1}{\sqrt{N}}\sum_l a_{\alpha,l} e^{i\mathbf{k}\cdot(\mathbf{R}_l + \mathbf{r}_\alpha)}$. Here, the operator $a^{(\dagger)}_{\alpha,l}$ annihilates (creates) a single particle in an atomic orbital $\alpha = A, B, C$ situated at position $\mathbf{r}_\alpha$ with respect to the position vector of a unit cell center $\mathbf{R}_l$, where $l = 1, 2, ..., N/3$ indexes unit cells. Due to the three sites per unit cell, we therefore have a three-band model. By inserting $a_{\alpha,l} = \frac{1}{2\pi\sqrt{N}}\sum_{\mathbf{k}} \tilde{a}_{\alpha,\mathbf{k}} e^{-i\mathbf{k}\cdot(\mathbf{R}_l + \mathbf{r}_\alpha)}$ into Eq. (2) in the main text, one obtains

$$H = \sum_{\mathbf{k};\alpha,\beta = A,B,C} H_{\alpha\beta}(\mathbf{k})\, \tilde{a}^{\dagger}_{\alpha,\mathbf{k}} \tilde{a}_{\beta,\mathbf{k}}. \quad (12)$$

Here, the Bloch Hamiltonian for the considered system on the kagome lattice, manifestly expressed in a real gauge, reads[48,49]

$$H(\mathbf{k}) = \begin{pmatrix} H_{AA}(\mathbf{k}) & H_{AB}(\mathbf{k}) & H_{AC}(\mathbf{k}) \\ H_{AB}(\mathbf{k}) & H_{BB}(\mathbf{k}) & H_{BC}(\mathbf{k}) \\ H_{AC}(\mathbf{k}) & H_{BC}(\mathbf{k}) & H_{CC}(\mathbf{k}) \end{pmatrix}, \quad (13)$$

with the corresponding (real) matrix elements, on setting $t = t' = t'' = -1$ and $\mathbf{k} = (k_1, k_2)$,

$$H_{AA}(\mathbf{k}) = -\mu + 2\cos(k_1), \quad (14)$$

$$H_{AB}(\mathbf{k}) = 2\cos(k_1/2 + k_2/2) + 2\cos(k_1/2 - k_2/2), \quad (15)$$

$$H_{AC}(\mathbf{k}) = 2\cos(k_2/2) + 2\cos(k_1 + k_2/2), \quad (16)$$

$$H_{BB}(\mathbf{k}) = -\mu + 2\cos(k_2), \quad (17)$$

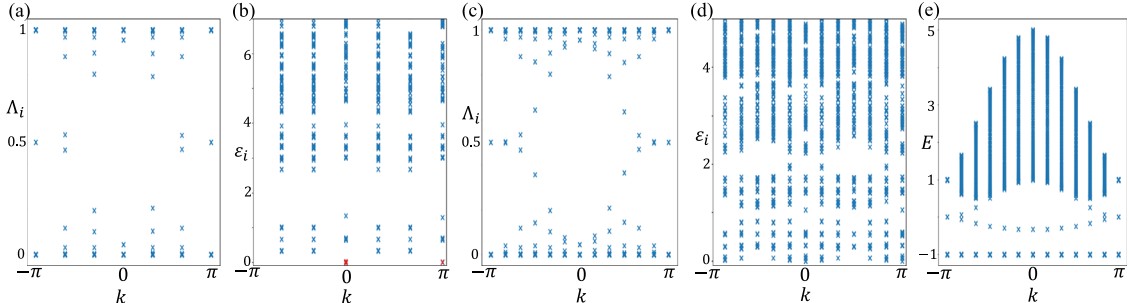

**Fig. 6 | Entanglement, one-body correlation, and physical spectra. a** One-body correlation spectrum $\Lambda_i$ on a thin torus with $L_y = 6$, **b** Many-body entanglement spectrum $\epsilon_i$ on an $L_y = 6$ torus. The red marker at $k = 0$ and $\epsilon_i = 0$ is the ground state of the partition 'A'. **c** and **d** are the one-body correlation spectrum $\Lambda_i$ and many-body entanglement spectrum $\epsilon_i$ for $L_y = 12$, respectively, to clarify the variation along $k$. **e** The physical spectrum $E$ on a cylinder of size $L_x = 120$ and $L_y = 12$ does not show an edge state with a spectral flow between the flat valence bands and dispersive conduction band.

$$H_{BC}(\mathbf{k}) = 2\cos(k_1/2) + 2\cos(k_1/2 + k_2), \tag{18}$$

$$H_{CC}(\mathbf{k}) = -\mu + 2\cos(k_1 + k_2). \tag{19}$$

We recognize that the Bloch Hamiltonian can be further rewritten as

$$H(\mathbf{k}) =$$
$$\begin{pmatrix} -\mu - 2 + 4\cos^2(k_1/2) & 4\cos(k_1/2)\cos(k_2/2) & 4\cos(k_1/2)\cos(k_1/2 + k_2/2) \\ 4\cos(k_1/2)\cos(k_2/2) & -\mu - 2 + 4\cos^2(k_2/2) & 4\cos(k_2/2)\cos(k_1/2 + k_2/2) \\ 4\cos(k_1/2)\cos(k_1/2 + k_2/2) & 4\cos(k_2/2)\cos(k_1/2 + k_2/2) & -\mu - 2 + 4\cos^2(k_1/2 + k_2/2) \end{pmatrix}. \tag{20}$$

or, more compactly,

$$H(\mathbf{k}) = (-\mu - 2)\mathbb{1}_3 + 4\mathbf{n}(\mathbf{k}) \otimes \mathbf{n}(\mathbf{k})^\mathsf{T}, \tag{21}$$

with $\mathbf{n}(\mathbf{k}) = \left(\cos(k_1/2), \cos(k_2/2), \cos(k_1/2 + k_2/2)\right)^\mathsf{T}$. Importantly, under such decomposition, the topology of the Euler bands in any three-band Hamiltonian satisfying a reality condition [$H(\mathbf{k}) = H^*(\mathbf{k})$] can be captured by the normalized vector $\hat{\mathbf{n}}(\mathbf{k}) = \mathbf{n}(\mathbf{k})/||\mathbf{n}(\mathbf{k})||$. In particular, in the considered model, the vector $\hat{\mathbf{n}}(\mathbf{k})$ reads

$$\hat{\mathbf{n}}(\mathbf{k}) = \frac{1}{\sqrt{\cos^2(k_1/2) + \cos^2(k_2/2) + \cos^2(k_1/2 + k_2)}}$$
$$\times \begin{pmatrix} \cos(k_1/2) \\ \cos(k_2/2) \\ \cos(k_1/2 + k_2/2) \end{pmatrix}, \tag{22}$$

and it fully determines the Euler curvature as

$$\mathrm{Eu} = \hat{\mathbf{n}} \cdot (\partial_{k_2}\hat{\mathbf{n}} \times \partial_{k_1}\hat{\mathbf{n}}). \tag{23}$$

The Euler curvature can be viewed as a skyrmion density in the momentum-space texture, with the skyrmion being spanned by $\hat{\mathbf{n}}$ over the Brillouin zone (BZ) square/torus. In particular, the Euler invariant is given by[37,48]

$$\chi = \frac{1}{2\pi}\int_{\mathrm{BZ}} \mathrm{d}^2\mathbf{k}\, \mathrm{Eu} = \frac{1}{2\pi}\int_{\mathrm{BZ}} \mathrm{d}^2\mathbf{k}\, \hat{\mathbf{n}} \cdot (\partial_{k_2}\hat{\mathbf{n}} \times \partial_{k_1}\hat{\mathbf{n}}) = 2Q, \tag{24}$$

and obtains $\chi = 1$ in the case of interest, which corresponds to the momentum-space meron (half-skyrmion) with the half-skyrmion number $Q = 1/2$[49]. Additionally, the vector $\mathbf{n}(\mathbf{k})$ fully captures the band dispersion present in the model, as Eq. (21) can be written as

$$H(\mathbf{k}) = (-\mu - 2)\mathbb{1}_3 + 4||\mathbf{n}(\mathbf{k})||^2 \hat{\mathbf{n}}(\mathbf{k}) \otimes \hat{\mathbf{n}}(\mathbf{k})^\mathsf{T}, \tag{25}$$

explicitly determining the band dispersion in the third band as $E_3(\mathbf{k}) = (-\mu - 2) + 4||\mathbf{n}(\mathbf{k})||^2$, contrary to the flat-band dispersion in the bottom Euler bands $E_1(\mathbf{k}) = E_2(\mathbf{k}) = (-\mu - 2)$. The band energies given by such dispersions manifestly have a gap across the entire Brillouin zone, as the norm of the vector $\mathbf{n}(\mathbf{k})$ is non-vanishing $||\mathbf{n}(\mathbf{k})|| > 0$ at every $\mathbf{k}$-point. This follows from the fact that the components of the vector $\mathbf{n}(\mathbf{k})$, $\cos(k_1/2)$, $\cos(k_2/2)$, and $\cos(k_1/2 + k_2/2)$, are not independent, with at least one of those terms being necessarily non-vanishing at any $\mathbf{k}$-point.

## Entanglement spectra of the non-interacting PEPS

Here we present the entanglement spectrum of the non-interacting kagome Euler model of the main text. For this purpose, we start with the momentum space Hamiltonian defined on a thin torus, i.e., $L_x \gg L_y$. In the insulating state, the bottom two flat bands are occupied and the dispersive conduction band is empty. We first write down the projector on the occupied state

$$\hat{P}(\mathbf{k}) = \sum_{i \in \text{occupied}} |\psi_i(\mathbf{k})\rangle\langle\psi_i(\mathbf{k})|. \tag{26}$$

The projector by definition has its eigenvalues restricted to 0 and 1. For the calculations performed on a lattice, we define the real space positions as $\mathbf{r} = n_1\mathbf{a}_1 + n_2\mathbf{a}_2$, where $n_{1(2)} \in \mathbb{Z}$ and $\mathbf{a}_{1(2)}$ are the lattice vectors. The corresponding reciprocal space momenta take the values as $\mathbf{k} = \frac{k_1}{2\pi}\mathbf{b}_1 + \frac{k_2}{2\pi}\mathbf{b}_2$, where $k_1, k_2 \in (-\pi, \pi]$ and $\mathbf{b}_{1(2)}$ are the reciprocal lattice vectors. For the kagome model here, we have chosen, $\mathbf{a}_1 = (\sqrt{3}/2, 1/2)$ and $\mathbf{a}_2 = (0, 1)$ as the lattice vectors. The corresponding reciprocal lattice vectors are $\mathbf{b}_1 = (4\pi/\sqrt{3}, 0)$ and $\mathbf{b}_2 = (-2\pi/\sqrt{3}, 2\pi)$. From the projector, we obtain a one-body correlation operator

$$G_{nm}(k_2) = \frac{1}{L_x}\sum_{k_1} e^{i2\pi k_1(n-m)}\hat{P}(k_1, k_2). \tag{27}$$

Since $G$ is also a projector, its eigenvalues are also restricted to 0 and 1. We partition the system into subsystems A and B, such that the entanglement spectrum between the two subsystems is given by the eigenvalues of the reduced density matrix $\rho_A$. The spectrum of the reduced density matrix $\rho_A$ can then be obtained from the spectrum of the reduced correlation matrix $G^A$ defined as[63]

$$G^A_{nm}(k_2) = G_{nm}(k_2); \quad n = 1, \dots L_x, m = 1, \dots, L_y. \tag{28}$$

In Fig. 6a and c, we show the spectrum of the reduced one-body correlation matrix $G^A$. The plots are obtained for system sizes $L_x = 120$ and $L_y = 6$ (12) for (a) and (c) respectively. The eigenvalues $\Lambda_i(k_2)$ of $G^A$

are bounded to lie in [0, 1], although, unlike the projector eigenvalues, they are not restricted to be 0 and 1. Indeed the in-gap eigenvalues are related to the topological Euler class of the model[64]. However, unlike the well-known case of Chern insulators, these in-gap modes in the one-body correlation spectrum of the Euler topology are not related to the physical edge states due to non-trivial topology, which typically has a spectral flow between the bulk conduction and valence bands. In Fig. 6(e) we explicitly show the absence of such topological edge states with a spectral flow between flat valence bands at $-1$ and dispersive conduction band. The physical energy spectrum is calculated for system size $L_x = 120$ and $L_y = 12$ with open boundary conditions along $L_x$ and periodic boundary conditions along $L_y$.

From the spectrum of $G^A$, we obtain the entanglement spectrum of the non-interacting model using the relation

$$\varepsilon(k_2) = - \sum_{i \in \text{occupied}} \log[\Lambda_i(k_2)]$$
$$- \sum_{j \in \text{unoccupied}} \log[1 - \Lambda_j(k_2)]. \tag{29}$$

To calculate the full many-body entanglement spectrum as shown in the figure, we first obtain the ground state of subsection $A$ by occupying 2/3 of the *highest* eigenvalues $\Lambda_i$, which is commensurate with the 2/3 filling of the whole system. Here, one should keep in mind that since the eigenvalues of the projector $\hat{P}$ of the occupied states lie at 1 and unoccupied states at 0, while constructing the ground state of $G^A$, one should start counting from $\Lambda_i \to 1$ as occupied, and going down in the eigenvalues $\Lambda_i$ corresponds to going up in the excitation spectrum. Once the ground state is identified, we obtain a many-body entanglement spectrum by creating excitations on this ground state. Since for the entanglement spectrum, we partition the system, while the filling fraction 2/3 is a constraint for the whole system, the subsystem A has eigenstates that have particle numbers different to 2/3 filling of the subsystem itself. Therefore to calculate the full many-body ground state, we first partition the subsystem A into different total particle number channels and from there create all possible particle-hole excitations. Then for each excited state configuration described by a fixed fermionic occupation number, we can obtain the entanglement spectrum using Eq. (29).

The many-body entanglement spectrum of the non-interacting model is shown in Fig. 6b and d for the system size $L_x = 120$ and $L_y = 6(12)$ respectively. We have taken the ground state entanglement energy to zero as a reference, which is shown by the red marker at $k = 0$ in Fig. 6b and d. Notice the presence of the zero entanglement energy state at $k = \pi$. This is obtained by considering a channel with one less (or more) particle in subsystem A than the exact 2/3 filling.

Comparing the many-body entanglement spectrum to the interacting case with $\alpha = 0$ in the main text (left panel in Fig. 5), we see a good agreement in their low energy features. In particular, in both cases, the ground state corresponds to $k = 0$ with the lowest entanglement energy. As we move to a finite $k$, the entanglement energy increases and eventually comes back down to the ground state value at $k = \pi$ creating a cusp-like feature in the low-energy entanglement spectrum. This low energy behavior can be traced back to the in-gap modes (around $\Lambda = 0.5$) in the one-body correlation spectrum shown in Fig. 6a and c. In the one-body correlation spectrum, for each mode near 0, there are two modes near 1, and therefore 2/3 filling corresponds to occupying all modes in the upper half of the correlation spectrum. The low energy excitations are then created near $\Lambda_i = 0.5$ with a very low energy cost, which leads to many low energy modes in the entanglement spectrum.

## Data availability
The datasets for the plots are available upon request.

## Code availability
The codes for our simulations are available upon request. They are based on the publicly available TeNPy library[60].

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

## Acknowledgements

We acknowledge funding from a New Investigator Award, EPSRC grant EP/W00187X/1, an EPSRC ERC underwrite grant EP/X025829/1, and a Royal Society exchange grant IES/R1/221060 as well as Trinity College, Cambridge (R.-J.S., G.C., and T.B.W.). This project was also supported by funding from the Rod Smallwood Studentship at Trinity College, Cambridge (W.J.J.).

## Author contributions

R.-J.S and T.B.W. initiated the project. T.B.W. constructed the PEPS and performed the numerical simulations with input on the geometry and specific models from R.-J.S, W.J.J., A.B and G.C. W.J.J. performed the momentum-space characterisation of the model and quantum geometry with inputs from R.J.S., T.B.W., and A.B. G.C. numerically benchmarked the entanglement of the free fermion spectra. All authors discussed the results. T.B.W. with input of R.-J.S. took the lead in the writing of the manuscript, but the final form of the manuscript benefitted from input from all authors.

## Competing interests

The authors declare no competing interests.
