## [Peer Review File · Nature Communications]

REVIEWER COMMENTS

Reviewer #1 (Remarks to the Author):

See attached file.

Reviewer #2 (Remarks to the Author):

Reviewer #3 (Remarks to the Author):

In the manuscript, the authors investigate the projected entangled pair states (PEPS) for a class of topological phases with Euler topology. They first construct the local gapped PEPS for the ground state of a noninteracting fermion model of Euler topology. Then, they generalize the PEPS and corresponding parent Hamiltonian to interacting cases using local unitary circuits. Finally, they demonstrate that the interacting PEPS share similar features in entanglement spectra with the noninteracting case, indicating the Euler topology for interacting models.

Previous studies showed that chiral topological states such as the ground state of a Chern insulator cannot be represented by a tensor network state with exponentially decaying correlation. This work studies the construction of tensor network states and parent Hamiltonians for a family of 2D gapped topological phase protected by crystalline symmetries and generalize the Euler band topology to interacting models for the first time to the best of my knowledge. Since the topological phases with Euler class has attracted growing attention due to the intriguing multi-gap topology and non-Abelian band nodes in recent years, we think that this work will advance our understanding of interacting topological phases with crystalline symmetries.

Overall, the results are interesting and valid and the manuscript is well written. We would like to recommend its publication in Nature Communications after the authors address the following issues.

1. For noninteracting fermion models, the Euler topology is derived from the homotopy of real vector bundles of occupied bands protected by crystalline symmetries like PT or C₂T symmetry. However, it is not very clear whether the symmetry could protect the Euler topology of PEPS in the presence of interactions. Can the authors comment on the symmetry protected topology of interacting PEPS?

2. We have a question about the possibility of local gapped PEPS representation of a 2D topological phase. Both Chern insulators and Euler insulators do not admit localized Wannier functions respecting the symmetry. Nevertheless, Euler insulators can be represented by local gapped PEPS while Chern insulators cannot. In the manuscript, the authors attribute the difference to the presence or absence of edge states which seems only applied to systems with open boundaries. Can the authors explain more about this?

3. In the paragraph titled “Free-fermion generalizations”, the authors state that “Whether this state has a non-zero Euler number depends on the specific choice of the β_i ”. It is not clear here how the coefficients β_i affect the topology of the PEPS. Can the authors add more explanations on this?

4. In page 4, there is a paragraph titled “Quantum geometry” discussing the quantum geometric properties of the noninteracting Euler bands, which seems to have little relation to other parts on PEPS in the main text. We suggest the authors to adjust the structure of manuscript or move the contents to Supplementary Information, which may improve the presentation of the paper.

5. In the construction of interacting PEPS of Euler topology, the authors introduce a parameter α in local unitary gates which parameterizes a family of interacting PEPS along with parent Hamiltonians. In Fig. 4, the entanglement spectra for $\alpha=\pi$ exhibits distinct behaviors from the small values of α resembling the noninteracting limit. Does this difference signify a transition from Euler topological phase to trivial phase at large value of α ? What is the relation between the value of α and the effective strength of interaction?

Reviewer #4 (Remarks to the Author):

Report: Exact projected entangled pair ground states with topological Euler invariant

September 9, 2024

In their manuscript, the authors provide an explicit construction of the PEPS representation for a two-dimensional topological state in the Euler class. After introducing a non-interacting model and deriving its exact solution, they propose a parametrized family of interacting models, which they briefly study through their entanglement properties.

This work is mathematically robust and offers an important contribution, as tensor network representations of two-dimensional topological systems remain a largely unexplored and open field. Finding a class of models that can be solved exactly, and that do not exhibit the power-law behavior associated with chiral PEPS (indicative of gapless systems), opens many promising avenues for future research.

However, some aspects of the manuscript are somewhat confusing and could benefit from a more pedagogical or concrete approach. Furthermore, the authors hint at the possibility of novel fractional models arising from the interplay of interactions and Euler topology. Providing a concrete example of such a model, along with its expected physical properties, would significantly strengthen the manuscript—particularly given the exotic nature of the Euler class.

In conclusion, this manuscript provides a solid foundation for further studies in the field, but certain aspects require clarification and expansion to make the results more impactful.

More detailed comments are given below.

Comments

1. **Introduction:** The authors mention that the lack of an exact finite PEPS for Chern insulators can be understood in terms of the presence of gapless one-dimensional edge modes. Is the existence of a PEPS in

the Euler class simply a consequence of the absence of such edge states? If so, is there any fundamental difference between the case studied in this manuscript and, for instance, the higher-order topological insulators whose PEPS representation is provided in Phys. Rev. B 101, 115134?

2. **Hamiltonian representation:** A diagram illustrating the Hamiltonian in Eq. (2) would be very helpful, even if placed in the Supplementary Material (SM). Similarly, the conventions used for the diagonalization in the SM are not fully specified and should be made explicit.
3. **Clarification on anticommutation relations:** (lines 123-129) The authors argue that the anticommutation relations $\{a_\alpha, a_\beta^\dagger\} = \delta_{\alpha,\beta} + \frac{1}{6}\delta_{\langle\alpha,\beta\rangle}$ imply that the Hamiltonian is not connected to an atomic insulator. While this is correct, further clarification on this point would be helpful to the reader.
4. **Projector M :** In Eq. (5), the authors introduce the projector M to project from the extended simplex space to the physical fermionic representation. Could the authors provide more detailed motivation for their choice of this particular projector?
5. **Uniqueness of the groundstate:** The argument justifying the equivalence between the PEPS state and the groundstate relies on the uniqueness of the latter at filling $2N/3$. While this is straightforward from the band theory perspective presented in the SM, a brief reminder of this argument in the main text before concluding the relevant paragraph would be appreciated.
6. **Reordering of paragraph (lines 140-150):** The paragraph from lines 140 to 150 would be more logically placed after the proof that the state is indeed a PEPS. Additionally, all tensors involved are fermionic, and it would be beneficial to underline this fact explicitly and refer to relevant papers that implement fermionic PEPS.
7. **Quantum geometry (lines 170 onward):** Starting around line 170, the authors briefly mention quantum geometry and note that the flat bands are "ideal." A more thorough introduction to the relevance of quantum geometry in this context would strengthen this section and provide better context for its importance.
8. **Entanglement spectrum and Euler class:** It is not immediately clear which special properties of the entanglement spectrum are characteristic of the Euler class. The graphs provided in the main text

do not highlight particularly standout features. Simply having a low-lying excitation at zero momentum does not seem exceptional. A more definitive "smoking gun" signature would strengthen the paper.

9. **Fractional phases and interactions:** In the conclusion, the authors speculate about the possibility of new fractional phases arising from interactions in the Euler class. The manuscript would be far more impactful if an explicit example of such a model or tensor network were provided. Additionally, given that the anomalous Hall effect typically implies the presence of chiral edge states, wouldn't this lead to a similar obstruction to PEPS as in the case of Chern insulators?

Summary of changes:

1. We moved the section on “Quantum geometry” to the Supplementary Information (SI).
2. We now give an example of a Gutzwiller-projected PEPS which is expected to be a fractional Euler insulator.
3. We added a figure illustrating the hoppings of the non-interacting Hamiltonian.
4. We provided further clarifications throughout the manuscript on the topological robustness of our Hamiltonians.

The specific changes to the manuscript can be gathered from the attached latexdiff-file.

Response to Reviewers #1 and #2

In their manuscript, the authors provide an explicit construction of the PEPS representation for a two-dimensional topological state in the Euler class. After introducing a non-interacting model and deriving its exact solution, they propose a parameterized family of interacting models, which they briefly study through their entanglement properties.

This work is mathematically robust and offers an important contribution, as tensor network representations of two-dimensional topological systems remain a largely unexplored and open field. Finding a class of models that can be solved exactly, and that do not exhibit the power-law behavior associated with chiral PEPS (indicative of gapless systems), opens many promising avenues for future research.

We cordially thank the Reviewer(s) for their careful reading of our manuscript and for their positive assessment of our work.

However, some aspects of the manuscript are somewhat confusing and could benefit from a more pedagogical or concrete approach. Furthermore, the authors hint at the possibility of novel fractional models arising from the interplay of interactions and Euler topology. Providing a concrete example of such a model, along with its expected physical properties, would significantly strengthen the manuscript—particularly given the exotic nature of the Euler class.

We thank the Referee for their points. We agree that the previous manuscript was not pedagogically optimally structured. We found the Reviewer’s points quoted further below very helpful in making the manuscript more accessible to a broad spectrum of readers. Regarding fractional Euler insulators, we now provide an explicit example of a projected entangled pair state (PEPS) with bond dimension $D = 4$ which is expected to be a quantum spin liquid Euler state in the same topological phase as fractional topologically ordered Euler insulators. We achieve this by applying a Gutzwiller projection on two PEPSs representing non-interacting Euler insulators. The resulting state is expected to have symmetry-enriched topological order with fractional statistics, similar to the Gutzwiller projection of two $p + ip$ superconducting states.

In conclusion, this manuscript provides a solid foundation for further studies in the field, but certain aspects require clarification and expansion to make the results more impactful.

We thank the Referee for their assessment that our manuscript provides a solid foundation for further studies in the field. Below, we provide detailed clarifications and expansions on the aspects raised by the Referee. We welcome these as an opportunity to further extend the scope and impact of our manuscript.

More detailed comments are given below.

Comments

1. Introduction: *The authors mention that the lack of an exact finite PEPS for Chern insulators can be understood in terms of the presence of gapless one-dimensional edge modes. Is the existence of a PEPS in the Euler class simply a consequence of the absence of such edge states? If so, is there any fundamental difference between the case studied in this manuscript and, for instance, the higher-order topological insulators whose PEPS representation is provided in Phys. Rev. B 101, 115134?*

We thank the Referee for this question. Indeed, the intuition is that systems that host gapless edge modes cannot be exactly captured by PEPS with a finite bond dimension and short-range correlations. The fact that we managed to exactly represent a gapped Euler state (which is topologically non-trivial, but lacks gapless edge modes) by a PEPS provides further support to his intuition. On the other hand, the higher-order topological insulators represented by PEPS in [Phys. Rev. B **101**, 115134 (2020)] have a bond dimension in one spatial direction that grows with the system size. We added a comment on this

aspect in the revised manuscript.

2. Hamiltonian representation: *A diagram illustrating the Hamiltonian in Eq. (2) would be very helpful, even if placed in the Supplementary Material (SM). Similarly, the conventions used for the diagonalization in the SM are not fully specified and should be made explicit.*

We thank the Referee for their helpful suggestions. We agree with the Reviewer that these points were not fully explained in the previous manuscript. Correspondingly, we added a figure illustrating the hoppings of the Hamiltonian (2) and further details on the diagonalization in the SM of the revised manuscript.

3. Clarification on anticommutation relations: *(lines 123-129) The authors argue that the anticommutation relations $\{a_\alpha, a_\beta^\dagger\} = \delta_{\alpha\beta} + \frac{1}{6}\delta_{(\alpha,\beta)}$ imply that the Hamiltonian is not connected to an atomic insulator. While this is correct, further clarification on this point would be helpful to the reader.*

We thank the Referee for their invaluable suggestion of the clarification. We agree that our previous explanation of this fact was not clear enough. We now argue that even if one deforms our quasiparticle operators a_\square , their non-trivial anticommutation properties are protected by $C_2\mathcal{T}$ symmetry. That is, the a_\square cannot be deformed into single-site operators (and similarly for the Hamiltonian) without breaking $C_2\mathcal{T}$ symmetry.

4. Projector M : *In Eq. (5), the authors introduce the projector M to project from the extended simplex space to the physical fermionic representation. Could the authors provide more detailed motivation for their choice of this particular projector?*

We thank the Referee for noticing that the choice of our projector M was not sufficiently motivated. The virtual (simplex) state is of the form $|\psi_v\rangle = \prod_\square c_\square |1_v 0_p\rangle = \prod_\square \frac{1}{\sqrt{6}} \sum_{i \in \square} c_i |1_v 0_p\rangle$, where the operators c_\square have *non-overlapping* support. The map $\hat{M}_j = a_j^\dagger c_j' c_j + c_j' - c_j$ has the following effect on the superposition of basis states in $|\psi_v\rangle$: If site $j = \square \cap \square'$ is neither affected by c_i with $i \in \square$ nor c_k with $k \in \square'$ (i.e., there are two virtual fermions at site j), then a physical fermion is created via a_j^\dagger . If site j is affected by either $c_{i \in \square}$ or $c_{k \in \square'}$ (i.e., there is one virtual fermion at site j), then no physical fermion is created at site j , leaving it in the physical vacuum state. If site j is affected by both $c_{i \in \square}$ and $c_{k \in \square'}$ (i.e., there is no virtual fermion at site j), the basis state is annihilated and not mapped to the physical space. This ensures that the final state is particle-number conserving. The negative sign in \hat{M}_j is necessitated by the fermionic anticommutation relations in the PEPS construction. We give details on these aspects after Eq. (5) in the revised manuscript.

5. Uniqueness of the groundstate: *The argument justifying the equivalence between the PEPS state and the groundstate relies on the uniqueness of the latter at filling $2N/3$. While this is straightforward from the band theory perspective presented in the SM, a brief reminder of this argument in the main text before concluding the relevant paragraph would be appreciated.*

We thank the Referee for helping us to improve the clarity of our line of arguments. For chemical potential $\mu = -2$, the ground state space S is the intersection of all null spaces of $P_\square = a_\square^\dagger a_\square$, i.e., $S = \text{span}\{\prod_\square a_\square a_\square^\dagger |n\rangle\}_n = \text{span}\{\prod_\square a_\square |n\rangle\}_n$, where $\{|n\rangle\}_n$ is a complete basis and we used that $P_\square^\perp = a_\square a_\square^\dagger$ is a projector onto the orthogonal complement of P_\square . Each a_\square eliminates one fermion from the basis state $|n\rangle$. Hence, the state with the highest occupation in S is $\prod_\square a_\square |1 \dots 1\rangle$, and it has filling $N - N/3 = 2N/3$. Any other state contained in S has a lower expectation value of the overall occupation. Once the chemical potential is increased above -2 , this highest occupation ground state becomes the unique ground state, as all other states contained in S have lower occupation expectation values and thus get penalized. We clarify these aspects in the main body of the revised manuscript.

6. Reordering of paragraph (lines 140-150): The paragraph from lines 140 to 150 would be more logically placed after the proof that the state is indeed a PEPS. Additionally, all tensors involved are fermionic, and it would be beneficial to underline this fact explicitly and refer to relevant papers that implement fermionic PEPS.

We thank the Referee for pointing this out. We have correspondingly moved the paragraph, which used to be located between lines 140 and 150 behind the proof that the PEPS is a ground state. Furthermore, we clarified that the underlying tensors are fermionic, citing the relevant literature.

7. Quantum geometry (lines 170 onward): Starting around line 170, the authors briefly mention quantum geometry and note that the flat bands are “ideal.” A more thorough introduction to the relevance of quantum geometry in this context would strengthen this section and provide better context for its importance.

We cordially thank the Referee for their further suggestion. As Reviewer #3 correctly pointed out:

“In page 4, there is a paragraph titled ‘Quantum geometry’ discussing the quantum geometric properties of the noninteracting Euler bands, which seems to have little relation to other parts on PEPS in the main text. We suggest the authors to adjust the structure of manuscript or move the contents to Supplementary Information, which may improve the presentation of the paper.”

That is, the section on quantum geometry is of too limited relevance to the central message of the manuscript to keep it in the main body of the paper. We have thus accordingly moved it to the Supplementary Information.

8. Entanglement spectrum and Euler class: It is not immediately clear which special properties of the entanglement spectrum are characteristic of the Euler class. The graphs provided in the main text do not highlight particularly standout features. Simply having a low-lying excitation at zero momentum does not seem exceptional. A more definitive “smoking gun” signature would strengthen the paper.

We thank the Referee for pointing out that the argument for Euler topology based on the entanglement spectrum was unclear in our previous manuscript. We agree that a single low-lying mode at momentum zero would have been a weak signature of the non-trivial topology. However, this mode joins *two* low-lying isolated branches of entanglement energies, symmetric around $K = 0$, which we did not convey in our previous manuscript. While this cusp is invisible, our text should have reflected that. We note that due to the non-trivial Euler class, our non-interacting model can be represented in terms of two degenerate Chern bands with opposite Chern numbers $C = \pm 1$ in a complexified basis [Nat. Phys. **16**, 1137–1143 (2020)]. Consequentially, due to the continuous connection to the non-interacting limit, our $K = 0$ mode in the interacting case is not isolated, but connected to two isolated branches (gapless counter-propagating modes). The analysis of the spectrum in the SI (and the connection to the free fermion case) show the existence of these two branches. Revealing these branches fully would require vastly larger system sizes, which are not accessible. We added a comment on this to the revised manuscript, making it more apparent that we are retrieving a “smoking gun” signature.

9. Fractional phases and interactions: In the conclusion, the authors speculate about the possibility of new fractional phases arising from interactions in the Euler class. The manuscript would be far more impactful if an explicit example of such a model or tensor network were provided. Additionally, given that the anomalous Hall effect typically implies the presence of chiral edge states, wouldn’t this lead to a similar obstruction to PEPS as in the case of Chern insulators?

We thank the Referee for this question, and we welcome it as an opportunity to expand our material.

We now provide an explicit example of a PEPS which is expected to be a fractional Euler insulator (see our point above). We agree with the Referee that chiral edge states would lead to similar obstructions to PEPS as in the case of Chern insulators, that is, spurious algebraically decaying correlations in the bulk. Since the above PEPS are obtained by a (local) Gutzwiller projection from the non-interacting, gapped Euler insulator PEPS, they will have exponentially decaying correlations and, therefore, not display emergent quantum anomalous Hall phenomena. However, if one adds interactions to our non-interacting Euler model Hamiltonian, such phenomena might arise and can be studied variationally with tensor network states, notwithstanding their algebraically decaying correlations [Phys. Rev. Lett. **111**, 236805 (2013)]. We stress that these facts underpin our results and open various avenues for future pursuits. Indeed, it shows that we discovered a class of exotic (projected entangled pair) systems in which such novel phenomena (new correlated gapped states and possible spin liquids) should be investigated. Evidently, this is the promising and deep direction that should be fleshed out in the near future, being an important reason why we believe this is a platform to publish our results.

Response to Reviewers #3 and #4

In the manuscript, the authors investigate the projected entangled pair states (PEPS) for a class of topological phases with Euler topology. They first construct the local gapped PEPS for the ground state of a noninteracting fermion model of Euler topology. Then, they generalize the PEPS and corresponding parent Hamiltonian to interacting cases using local unitary circuits. Finally, they demonstrate that the interacting PEPS share similar features in entanglement spectra with the noninteracting case, indicating the Euler topology for interacting models.

Previous studies showed that chiral topological states such as the ground state of a Chern insulator cannot be represented by a tensor network state with exponentially decaying correlation. This work studies the construction of tensor network states and parent Hamiltonians for a family of 2D gapped topological phase protected by crystalline symmetries and generalize the Euler band topology to interacting models for the first time to the best of my knowledge. Since the topological phases with Euler class has attracted growing attention due to the intriguing multi-gap topology and non-Abelian band nodes in recent years, we think that this work will advance our understanding of interacting topological phases with crystalline symmetries.

Overall, the results are interesting and valid and the manuscript is well written. We would like to recommend its publication in Nature Communications after the authors address the following issues.

We thank the Reviewer(s) for their careful reading of our manuscript and appreciate their splendid summary and positive assessment of our work.

1. For noninteracting fermion models, the Euler topology is derived from the homotopy of real vector bundles of occupied bands protected by crystalline symmetries like \mathcal{PT} or $\mathcal{C}_2\mathcal{T}$ symmetry. However, it is not very clear whether the symmetry could protect the Euler topology of PEPS in the presence of interactions. Can the authors comment on the symmetry protected topology of interacting PEPS?

We agree with the Referee that the protection of Euler topology under $\mathcal{C}_2\mathcal{T}$ symmetry in the presence of interactions is an open question. In our case, we can guarantee that the symmetry-protected Euler topology is preserved, as our interacting PEPS and interacting model Hamiltonian are both related via a symmetry-preserving shallow quantum circuit to their non-interacting counterparts. A shallow quantum circuit does not change topological features, as these are global. We clarify these aspects in the revised manuscript.

2. We have a question about the possibility of local gapped PEPS representation of a 2D topological

phase. Both Chern insulators and Euler insulators do not admit localized Wannier functions respecting the symmetry. Nevertheless, Euler insulators can be represented by local gapped PEPS while Chern insulators cannot. In the manuscript, the authors attribute the difference to the presence or absence of edge states which seems only applied to systems with open boundaries. Can the authors explain more about this?

This is an important point raised by the Referee, which we clarified in the revised manuscript: Chern insulators do not admit any (exponentially) localized Wannier function basis (in any gauge), and Euler insulators do not admit localized Wannier functions, in any gauge respecting both $C_2\mathcal{T}$ symmetry and the three-band (gapped) conditions (i.e., Wannierization assuming a partitioning into a two-band topological subspace with the Euler invariant, which is isolated from a higher-energy trivial band by a gap across the entire Brillouin zone). Therefore, there is no short-range correlated PEPS representing a Chern insulator (independently of any considered/chosen gauge), whereas there can be short-range correlated PEPS representing Euler insulators if $C_2\mathcal{T}$ symmetry is broken (specifically by the gauge choice), or the three-bands condition is violated, *in the PEPS construction itself*. Our PEPS fall into the later category (as mentioned in the revised main text; given the gauge used), as the virtual state possesses two fermions per site, that is, six fermions per unit cell, whereas the physical state only possesses three fermions per unit cell.

3. In the paragraph titled “Free-fermion generalizations”, the authors state that “Whether this state has a non-zero Euler number depends on the specific choice of the β_i ”. It is not clear here how the coefficients β_i affect the topology of the PEPS. Can the authors add more explanations on this?

We thank the Referee for pointing this out. After careful consideration, we became aware that for any permissible choice of $\{\beta_i\}$, the corresponding PEPS is an Euler insulator, as the corresponding parent Hamiltonians are always gapped and continuously connected to the original one. We corrected this in the revised manuscript.

4. In page 4, there is a paragraph titled “Quantum geometry” discussing the quantum geometric properties of the noninteracting Euler bands, which seems to have little relation to other parts on PEPS in the main text. We suggest the authors to adjust the structure of manuscript or move the contents to Supplementary Information, which may improve the presentation of the paper.

We have followed the Referee’s suggestion and moved the paragraph on “Quantum geometry” to the Supplementary Information, improving the presentation and coherence of our manuscript.

5. In the construction of interacting PEPS of Euler topology, the authors introduce a parameter α in local unitary gates which parameterizes a family of interacting PEPS along with parent Hamiltonians. In Fig. 4, the entanglement spectra for $\alpha = \pi$ exhibits distinct behaviors from the small values of α resembling the noninteracting limit. Does this difference signify a transition from Euler topological phase to trivial phase at large value of α ? What is the relation between the value of α and the effective strength of interaction?

We thank the Referee for this important point. α is a parameter in the shallow quantum circuit we apply on our non-interacting PEPS with Euler topology. As topological features are invariant under the application of symmetry-preserving shallow quantum circuits, the PEPS retains its topological features for all $\alpha \in [0, \pi]$, i.e., no transition to the trivial phase can occur. Similarly, the interacting Hamiltonian is related via the shallow quantum circuit to the non-interacting one, i.e., the gap (in fact, the entire spectrum) remains invariant as α is increased. The interacting Hamiltonian as a function of α is given in Eq. (12) and contains many-body interactions. The four-body interactions have an amplitude of order $\mathcal{O}(\alpha)$ (i.e., the Hubbard-like effective interaction strengths are linear in α), and the higher-body interactions are of higher order. We clarified these aspects in the revised manuscript.

REVIEWERS' COMMENTS

Reviewer #1 (Remarks to the Author):

The authors have done a very thorough job at taking all remarks by the referees into account when revising the manuscript. The answers to all questions are precise and right to the point. As far as I am concerned, the manuscript can be published in its present form.

Please see attached file for a final minor point to be clarified.

Reviewer #2 (Remarks to the Author):

I co-reviewed this manuscript with one of the reviewers who provided the listed reports. This is part of the Nature Communications initiative to facilitate training in peer review and to provide appropriate recognition for Early Career Researchers who co-review manuscripts. Ψ

Reviewer #3 (Remarks to the Author):

The authors have addressed all our comments raised in the first report and made substantial improvements to the manuscript. The authors have clarified details in construction of PEPS and nontrivial topology of interacting Euler states obtained from quantum circuits. In particular, the authors give an example of PEPS of an interacting Euler state which may have symmetry-enriched topological order. We appreciate the authors for their efforts of revising the manuscript and replying to all referees' comments, which convince us of the validness and novelty of this paper. Essentially, this work shows the possibility of tensor network representation of topological phases beyond chiral topological states focused on before. We will recommend the paper for publication after the authors address the following comments.

Minor comment:

In the revised manuscript, the authors apply a Gutzwiller projection on two copies of PEPS of noninteracting Euler insulator and then claim that this phase realizes a topological state named Euler quantum spin liquid in analogous to the chiral spin liquid obtained from projected $p+ip$

superconducting states studied previously. Here, the description of this interacting topological state is not very clear. We hope the authors to explain more about the distinct topological properties of this phase, such as its fractional statistics, which will make the manuscript more accessible to readers.

Reviewer #4 (Remarks to the Author):

Second Report: Exact projected entangled pair ground states with topological Euler invariant

November 23, 2024

In their reply, the authors provided satisfactory answers to all comments I raised in my previous report. Consequently, and given my evaluation of the quality of their work, I strongly recommend publication of their article in your journal. I will nonetheless add one minor question that I would be interested in seeing addressed.

Comments

1. The authors mention in their reply that the key reasons why they can build an exact PEPS with a finite bond dimension in the Euler class is the use of six bands in the construction of the tensor network. At the same time, they project out these bands locally, obtaining a standard fermionic PEPS with the correct physical degrees of freedom. As such, there seem to be an apparent contradiction in this statement. Could the authors clarify this point?

Summary of changes:

1. We added a comment on why we obtain a PEPS of finite bond dimension representing an Euler insulator at the end of the subsection "Euler class".
2. We further clarify the topological features of the Gutzwiller-projected non-interacting Euler states at the top of page 5.
3. We carried out minor changes throughout the manuscript in order to conform with the Nature Communications format guidelines.

Response to Reviewers #1 and #2

The authors have done a very thorough job at taking all remarks by the referees into account when revising the manuscript. The answers to all questions are precise and right to the point. As far as I am concerned, the manuscript can be published in its present form.

We cordially thank the Reviewer(s) for their careful reading of our revised manuscript and for their positive assessment of our work.

In their reply, the authors provided satisfactory answers to all comments I raised in my previous report. Consequently, and given my evaluation of the quality of their work, I strongly recommend publication of their article in your journal. I will nonetheless add one minor question that I would be interested in seeing addressed.

Comments

1. The authors mention in their reply that the key reasons why they can build an exact PEPS with a finite bond dimension in the Euler class is the use of six bands in the construction of the tensor network. At the same time, they project out these bands locally, obtaining a standard fermionic PEPS with the correct physical degrees of freedom. As such, there seem to be an apparent contradiction in this statement. Could the authors clarify this point?

We thank the Referee for this question. Crucially, the PEPS construction breaks the gauge symmetry protecting the topological phase. This is made possible by starting with six virtual fermions per unit cell, breaking the 3-band condition. While the final state indeed corresponds to a three-band Hamiltonian, it is obtained from a gauge-symmetry breaking initial state via a projection, i.e., a non-invertible map (as opposed to a unitary quantum circuit). Hence, the PEPS is based on a gauge-symmetry breaking map (connecting states with different symmetries), which allows it to exactly represent a non-interacting symmetry-protected topological state. We clarify this aspect in the revised manuscript.

Response to Reviewers #3 and #4

The authors have addressed all our comments raised in the first report and made substantial improvements to the manuscript. The authors have clarified details in construction of PEPS and nontrivial topology of interacting Euler states obtained from quantum circuits. In particular, the authors give an example of PEPS of an interacting Euler state which may have symmetry-enriched topological order. We appreciate the authors for their efforts of revising the manuscript and replying to all referees' comments,

which convince us of the validness and novelty of this paper. Essentially, this work shows the possibility of tensor network representation of topological phases beyond chiral topological states focused on before. We will recommend the paper for publication after the authors address the following comments.

We thank the Reviewer(s) for their careful reading of our revised manuscript and recommending publication of our work.

Minor comment:

In the revised manuscript, the authors apply a Gutzwiller projection on two copies of PEPS of noninteracting Euler insulator and then claim that this phase realizes a topological state named Euler quantum spin liquid in analogous to the chiral spin liquid obtained from projected $p+ip$ superconducting states studied previously. Here, the description of this interacting topological state is not very clear. We hope the authors to explain more about the distinct topological properties of this phase, such as its fractional statistics, which will make the manuscript more accessible to readers.

We thank the referee for pointing this out. In the revised manuscript, we further elucidate the topological features of the Gutzwiller-projected Euler state: As the Gutzwiller projection preserves the $\mathcal{C}_2\mathcal{T}$ symmetry and restores the constraint of Hilbert space dimension 2 per lattice site, the final state will be in the same symmetry class as the original Euler insulator but have spin degrees of freedom. In analogy to the Gutzwiller projection of two $p + ip$ states, we expect our final state to be topologically ordered yet also protected by the same symmetries as the Euler insulator. We thus conjecture that our Gutzwiller-projected state is a symmetry-enriched topologically ordered state. As such, it would have anyonic excitations and topological ground state degeneracy on the torus, even if the protecting symmetry is broken. If the protecting symmetry is not broken, additional features of non-interacting Euler insulators are expected, such as the cusp in the entanglement spectrum at $K = 0$ that we describe in our work. We further elaborate on these aspects in our new manuscript.